# Protection against Oxidative Stress and Metabolic Alterations by Synthetic Peptides Derived from *Erythrina edulis* Seed Protein

**DOI:** 10.3390/antiox11112101

**Published:** 2022-10-25

**Authors:** Nathaly Rodríguez-Arana, Karim Jiménez-Aliaga, Arturo Intiquilla, José A. León, Eduardo Flores, Amparo Iris Zavaleta, Víctor Izaguirre, Christian Solis-Calero, Blanca Hernández-Ledesma

**Affiliations:** 1Laboratorio de Biología Molecular, Grupo de Investigación BIOMIAS, Facultad de Farmacia y Bioquímica, Universidad Nacional Mayor de San Marcos, Jr. Puno N° 1002, Lima 4559, Peru; 2Departamento de Ciencia de los Alimentos y Tecnología Química, Facultad de Ciencias Químicas y Farmacéuticas, Universidad de Chile, Santos Dumont 964, Independencia, Santiago 8380494, Chile; 3Department of Bioactivity and Food Analysis, Institute of Food Science Research (CIAL, CSIC-UAM, CEI UAM+CSIC), Nicolás Cabrera 9, 28049 Madrid, Spain

**Keywords:** legume proteins, *Erythrina edulis*, multifunctional peptides, antioxidant activity, molecular docking

## Abstract

The ability of multifunctional food-derived peptides to act on different body targets make them promising alternatives in the prevention/management of chronic disorders. The potential of *Erythrina edulis* (pajuro) protein as a source of multifunctional peptides was proven. Fourteen selected synthetic peptides identified in an alcalase hydrolyzate from pajuro protein showed in vitro antioxidant, anti-hypertensive, anti-diabetic, and/or anti-obesity effects. The radical scavenging properties of the peptides could be responsible for the potent protective effects observed against the oxidative damage caused by FeSO_4_ in neuroblastoma cells. Moreover, their affinity towards the binding cavity of angiotensin-converting enzyme (ACE) and dipeptidyl peptidase IV (DPP-IV) were predicted by molecular modeling. The results demonstrated that some peptides such as YPSY exhibited promising binding at both enzymes, supporting the role of pajuro protein as a novel ingredient of functional foods or nutraceuticals for prevention/management of oxidative stress, hypertension, and metabolic-alteration-associated chronic diseases.

## 1. Introduction

In recent years, the global incidence and mortality of non-communicable diseases (NCDs), such as cardiovascular and neurodegenerative disorders, diabetes, and cancer, have increased exponentially, accounting for over 70% of annual deaths worldwide [1]. High saturated fat and sugar content, physical inactivity, insufficient sleep, intense psychological stress, low sun exposure, environmental contamination, and smoking or alcohol abuse are recognized as the main characteristics of the Western diet that increase the risk of NCDs. On the contrary, a healthy diet containing functional foods has been associated with a lower risk of several of these disorders [2,3]. All NCDs share a common etiological characteristic, currently known as metaflammation, which is considered as a subclinical and permanent inflammation. As a result of this status, a metabolic cascade involving cell oxidative stress, atherosclerotic processes, and insulin resistance occurs, which gradually generates a substantial deterioration in the organism [4].

Reactive oxygen species (ROS) are essential components of living cells resulting from normal physiological processes and xenobiotic detoxification. However, ROS are responsible for the oxidation of lipids and proteins and the consequent production of toxic compounds such as malondialdehyde (MDA) and protein carbonyls. The accumulation of ROS exceeding the tolerance of the cellular antioxidant defenses results in an oxidative stress status involved in development and progression of multiple NCDs [5,6]. Moreover, persistent exposure of pancreatic β-cells to ROS provokes their dysfunction and induced insulin resistance resulting in metabolic-alteration-associated disorders. Similarly, the role of ROS in the initiation, development, and clinical outcomes of cardiovascular diseases is already known, and different mechanisms of antioxidant action in the prevention of these disorders have been proposed [4]. In relation to the pathophysiological process of hypertension, it has been described that angiotensin II is involved in alterations of the oxidative balance, stimulating the production of ROS, which produces excessive oxidation of macromolecules, damaging cardiovascular structures [7].

Although changes in dietary habits are still recognized as crucial, in the last years, many authors are suggesting the importance of the intake of antioxidants in the prevention of oxidative-stress-associated diseases, especially when they are taken within a normal diet [8]. In addition, the potential of multifunctional antioxidant compounds exerting their mechanism of action on blood pressure and/or metabolic risk factors is highlighted to reduce the risk of most of NCDs. Among them, multifunctional peptides, able to exert more than one biological effect through their action on several targets, represent a promising area with multiple applications [9]. In comparison with monofunctional peptides, they may be recognized as an enhancement due to reduced adverse side effects and costs [10]. In the last years, the interest has focused on multifunctional peptides derived from legume seed proteins [11,12]. Peptide KTYGL, identified in *Phaseolus vulgaris* seeds, has been characterized by its potent antioxidant, and angiotensin-converting enzyme (ACE) and dipeptidyl peptidase IV (DPP-IV)-inhibitory activities [13]. Other peptides contained in the same seed have been also reported to act as α-amylase and pancreatic lipase inhibitors [14]. Similarly, lunasin contained in soybean protein is recognized as an important multifunctional peptide with antioxidant, chemopreventive, hypocholesterolemic, anti-hypertensive, and nervous system modulatory activities [15]. Current research is focused on the potential of new legume species as source of multifunctional peptides.

*Erythrina edulis* (pajuro) is a legume that grows wild in the inter-Andean valleys and jungle of Peru. Its seeds have been traditionally used as a staple food due to its high nutritional value, with an important content of high quality and digestibility proteins (18–25%), carbohydrates (51%), minerals (phosphorus, iron, sulfur, sodium, potassium, manganese, and calcium), and vitamins (vitamin C, thiamin, niacin, and riboflavin) [16]. The regular consumption of this legume has also been associated to health benefits as diuretic, hypotonic, and osteoporosis prophylactic due to its content in phytochemicals such as saponins, alkaloids, flavonoids, and polyphenols [17]. Although the protein content is high, the information about their potential as source of bioactive peptides is still limited. Our first studies focused on the radical scavenging capacity of an alcalase hydrolyzate from pajuro proteins [18]. Fourteen peptides were identified as potentially responsible for the observed effects by an in silico analysis using the BIOPEP database. Moreover, their sequence suggest that these peptides could exert other properties making them promising preventive compounds against oxidative stress and metabolic-disfunction-associated diseases. Thus, the objective of the present article was to evaluate the multifunctional properties of synthetic peptides derived from pajuro proteins to determine their real contribution on the effects exerted by the pajuro hydrolyzate. The in vitro antioxidant, anti-hypertensive, anti-diabetic, and anti-obesity activities were determined, and the mechanism of action on blood pressure and metabolic biomarkers was deciphered through molecular docking. Moreover, a FeSO_4_-induced cell model was used to confirm the neuroprotective effects of synthetic peptides.

## 2. Materials and Methods

### 2.1. Materials

The 2,2′-azino-bis(3-ethylbenzothiazoline-6-sulfonic acid) (ABTS), 6-hydroxy-2,5,7,8-tetramethylchromium-2-carboxylic acid (Trolox), fluorescein disodium (FL), 2,2′-azobis(2-amidinopropane dihydrochloride) (AAPH), ACE from rabbit lung, hippuryl-histidyl-leucine (HHL), captopril, α-amylase type VI-B from porcine pancreas, 4-nitrophenyl α-D-glucopyranoside, acarbose, α-glucosidase from *Saccharomyces cerevisiae*, DPP-IV inhibitor detection kit MAK203, sitagliptin, porcine pancreatic lipase type II, sodium deoxycholate, orlistat, 4-nitrophenyl palmitate, RPMI-1640 medium, fetal bovine serum (FBS), L-glutamine solution, sodium pyruvate, non-essential amino acids (NEAA), gentamicin, 3-[4,5-dimethylthiazol-2-yl]-2,3-tetrazolium bromide (MTT), 2′-7′dichlorofluorescin diacetate (DCFA-DA), β-nicotinamide adenine dinucleotide (NADH), MAK094E carbonyl compound kit, and bovine serum albumin (BSA) were purchased from Sigma-Aldrich (St. Louis, MO, USA). Iron II sulfate heptahydrate (Fe_2_SO_4_.7H_2_O), 3,5-dinitrosalicylic acid, and thiobarbituric acid (TBA) were purchased from Merck Millipore Corp (Darmstadt, Germany). All other reagents were of analytical grade.

### 2.2. Obtention of Synthetic Peptides from Pajuro

Peptides (DGLGYY, AALWE, MFTGPY, YDLHGY, GESWCR, CCGDYY, NGENDWR, TWVV, GSYHDSK, YYLTR, SQLPGW, GPPW, YPSY, SKDAPY) were synthesized by the conventional Fmoc solid-phase synthesis and its purity was determined by HPLC-MS analysis, obtaining values higher than 98%.

### 2.3. Antioxidant Activity of Synthetic Pajuro-Derived Peptides

The ABTS^•+^ scavenging activity was determined as previously reported by Re et al. [19]. A volume of 20 μL of either PBS (blank), Trolox (2–25 µM) (standard), or sample was mixed with 980 μL of diluted ABTS^•+^ solution, and the absorbance was measured at 734 nm after 7 min incubation at room temperature in an Infinite M200 Pro plate reader (Tecan Group AG, Männendorf, Switzerland). To calculate the Trolox equivalent antioxidant capacity (TEAC), expressed as µmol of Trolox equivalents (TE)/µmol peptide, the gradient of the plot of the percentage inhibition versus concentration (for the peptide) was divided by the gradient of the plot obtained for Trolox. Three independent runs were performed for each sample.

The oxygen radical absorbance capacity (ORAC) was determined according to the protocol described by Hernández-Ledesma et al. [20]. The reaction mixture contained 30 nM FL, 12 mM AAPH, Trolox (0–5 nM), or sample (at different concentrations) in 75 mM phosphate buffer (pH 7.4). It was incubated for 120 min at 37 °C, and the fluorescence was recorded every 2 min in an Infinite M200 Pro plate reader (Tecan Group AG) at λ_excitation_ and λ_emission_ of 485 and 520 nm, respectively. The Icontrol software version 1.11.10 was used to control the equipment and measurement of the fluorescence, and 96-well microplates (96F untreated, pure Grade™, Brand, Wertheim, Germany) were used. Three independent runs were performed for each sample. ORAC values were expressed as μmol TE/μmol peptide.

### 2.4. Angiotensin-Converting Enzyme (ACE) Inhibitory Activity

The in vitro ACE inhibitory activity of peptides was measured following the methodology described by Hayakari et al. [21], with some modifications. The reaction mixture (200 µL) contained 1 mU ACE, 1.25 mM HHL, and sample or captopril (used as standard) at different concentrations. It was incubated at 37 °C for 60 min. After inactivating the enzyme by heating at 100 °C for 10 min, 100 µL of a 3% solution of trichloro-5-triazine/dioxan was added, the mixture was shaken and centrifuged at 1000× *g* for 10 min, and the absorbance was measured at 382 nm in the Infinite M200 Pro plate reader (Tecan Group AG). Three independent runs were performed for each sample. The activity was expressed as IC_50_ or peptide concentration required to inhibit the activity by 50%.

### 2.5. In Vitro Anti-Diabetic Activity

The α-amylase inhibition assay was carried out following the methods described by Subramanian et al. and Kumar et al. [22,23], with some modifications. Then, 200 µL of mixture, containing α-amylase (800 mU), 0.125% starch, and peptide (100 µM) or standard (at different concentrations) in 0.1 M phosphate buffer (pH 6.9), was incubated at 37 °C for 30 min. After adding 100 µL of a solution containing 1% 3,5-dinitrosalicylic acid and 30% sodium potassium tartrate in 0.4 M NaOH, shaking, and incubating at 100 °C for 10 min, the absorbance was measured at 540 nm in an Infinite M200 Pro plate reader (Tecan Group AG). Three independent runs were performed for each sample. The results were expressed as percentage of the negative control, considered as 100%.

The α-glucosidase inhibitory activity was determined as previously reported [24,25], with some modifications. The reaction mixture contained 10 mU α-glucosidase, 5 mM 4-nitrophenyl α-D glucopyranoside, and peptide (100 µM) or standard acarbose (at different concentrations) in 0.1 M phosphate buffer (pH 6.9) and was incubated at 25 °C for 30 min, measuring the absorbance each 2 min at 405 nm in the Infinite M200 Pro plate reader (Tecan Group AG). Three independent runs were performed for each sample. The results were expressed as percentage of the negative control, considered as 100%.

The DPP-IV inhibitory activity was determined using the DPP-IV Inhibitor Screening Kit MAK203, according to the manufacturer’s instructions. Then, 100 µL of the reaction mixture containing 5 µL of enzymatic solution, peptide (100 µM) or standard sitagliptin, (at different concentrations) and 5 µL of substrate was incubated at 37 °C for 30 min. After this time, the fluorescence was measured each 2 min at λ_excitation_ and λ_emission_ of 360 and 460 nm, respectively, in the Infinite M200 Pro plate reader (Tecan Group AG). Three independent runs were performed for each sample. The results were expressed as IC_50_ value.

### 2.6. Pancreatic Lipase Inhibitory Activity

The pancreatic lipase inhibitory activity was determined following the protocols previously described [26,27], with some modifications. Then, 200 µL of mixture containing 50 mM sodium deoxycholate, 0.2 mM CaCl_2_, 12 mU type 2 pancreatic lipase, peptide (100 µM) or standard orlistat, and 0.125 mM substrate 4-nitrophenyl palmitate in 50 mM phosphate buffer (pH 8.0) was incubated at 37 °C for 60 min, measuring the absorbance each 2 min at 405 nm in the Infinite M200 Pro plate reader (Tecan Group AG). Three independent runs were performed for each sample. The results were expressed as percentage of the negative control, considered as 100%.

### 2.7. Protective Effects of Synthetic Pajuro-Derived Peptides in SH-SY5Y Cells

Human neuroblastoma SH-SY5Y cells were obtained from the American Type Culture Collection (ATCC, HTB-38, Rockville, MD, USA) and grown in RPMI-1640 medium supplemented with 10% FBS, 2 mM L-glutamine, 1 mM sodium pyruvate, 1 mM NEAA, and 50 μg/mL gentamicin in a humidified incubator containing 5% CO_2_ and 95% air at 37 °C.

#### 2.7.1. Effects on Cell Viability

The effects of pajuro-derived peptides on cell viability were determined using the MTT assay, following the protocol previously described by Palma-Albino et al. [28]. SH-SY5Y cells were seeded (2 × 10^4^ cells/well) onto 96-well plates in complete medium with 1% FBS and incubated at 37 °C for 16 h. After discarding the culture medium, the peptide sample was added (at concentration of 10, 25, and 100 µM), and the plate was incubated for 24 h. Upon removing the supernatant, a MTT solution (2 mg/mL) was added, and the plate was incubated for 60 min at 37 °C. After this time, the supernatant was discarded, and the formazan crystals were dissolved in dimetilsulfoxide (DMSO). The absorbance was measured at 570 nm in the Infinite M200 Pro plate reader (Tecan Group AG). Three independent runs were performed for each sample. The results were expressed as percentage of the control, considered as 100%.

#### 2.7.2. Protective Effects against Oxidative Stress Induced by FeSO_4_

Intracellular ROS levels were quantified as previously described by LeBel et al. [29]. SH-SY5Y cells were seeded onto 96-well plates (2 × 10^4^ cells/well) in complete medium with 1% FBS and incubated for 16 h at 37 °C. Once aspirated the medium, cells were washed with PBS, and incubated with 100 µM of each peptide, dissolved in complete medium with 1% FBS, for 3 h at 37 °C. After discarding the peptide treatment, 100 µL of a solution containing 10 mM DCFA-DA and 1 M glucose in PBS was added to the wells, and the plate was incubated at 37 °C for 30 min. The supernatant was removed, and after washing the cells with PBS, a solution of 200 µM FeSO_4_ was added, measuring the fluorescence after 60 min at excitation and emission wavelengths of 485 and 530 nm, respectively, in the Infinite M200 Pro plate reader (Tecan Group AG). Three independent runs were performed for each sample. The results were expressed as ROS levels (% compared with the control, considered as 100%).

#### 2.7.3. Determination of Thiobarbituric Acid Reaction Substances (TBARS)

The thiobarbituric acid reactive substances (TBARS) content has been used as an index of lipid peroxidation. The assay was performed following the method described by Jimenez-Aliaga et al. [30], with some modifications. SH-SY5Y cells were seeded onto 60 mm dishes (7.5 × 10^5^ cells/dish) in complete medium with 10% FBS and incubated for 72 h. Once discarded the supernatant, cells were treated with peptides at 100 µM in complete medium with 1% FBS for 3 h at 37 °C. Then, cells were washed with PBS and challenged with 200 µM FeSO_4_ in complete medium with 1% FBS. After 1 h induction, supernatant was collected and stored. Then, cells were washed with PBS, pellets were collected, and washed again. The collected pellet was dissolved in 50 mM phosphate buffer (pH 7.4) and ultrasonicated. Then, 30 μL of the cellular suspension was mixed with 250 μL of 1% phosphoric acid and 75 μL of TBA, and the mixture was incubated at 100 °C for 45 min. Then, the mixture was cooled down, centrifuged at 3000× *g* for 5 min at 4 °C, and the fluorescence was measured (485 and 530 nm as excitation and emission wavelengths, respectively) in the Infinite M200 Pro plate reader (Tecan Group AG). A standard curve with MDA was used. Three independent runs were performed for each sample. The results were expressed as nmol MDA/mg protein.

#### 2.7.4. Determination of Lactate Dehydrogenase (LDH) Activity

Here, 100 µL of the reaction solution containing 0.18 mM sodium pyruvate and 0.6 mM NADH in phosphate buffer (50 mM, pH 7.4) was mixed with 100 µL of the supernatant previously collected (Section 2.7.3), and incubated for 2 min. Then, the fluorescence was measured at λ_excitation_ and λ_emission_ of 360 and 460 nm, respectively, in the Infinite M200 Pro plate reader (Tecan Group AG). Three independent runs were performed for each sample. The results were expressed as LDH activity (% compared with the control, considered as 100%).

#### 2.7.5. Determination of Carbonyl Compounds

SH-SY5Y cells were seeded onto 60 mm dishes (7.5 × 10^5^ cells/dish) in complete medium and incubated for 72 h. Once the medium was removed, cells were incubated with peptides (25, 50, and 100 µM) dissolved in complete medium with 1% FBS for 3 h at 37 °C. After washing with PBS, cells were induced with 200 µM FeSO_4_ for 1 h. Once washed twice with PBS, cells were collected by scraping, resuspended in 100 μL of 2,4-dinitrophenylhydrazine (DNPH), and incubated for 10 min at 25 °C. Then, 30 μL of trichloroacetic acid (TCA) was added, and the solution was shaken, incubated on ice for 5 min, and centrifuged at 13,000× *g* for 2 min. After discarding the supernatant, the pellet was resuspended in 500 μL of ice-cold acetone, sonicated for 30 s, and incubated at −20 °C for 5 min. The solution was centrifuged at 13,000× *g* for 2 min, and the supernatant was removed. Finally, 200 μL of guanidine was added and sonicated for 10 s. Next, 100 μL were transferred to a 96-well plate and the absorbance was measured at 375 nm in the Infinite M200 Pro plate reader (Tecan Group AG). A calibration curve of an accurately prepared standard BSA solution (8–100 mg/mL) was also run for quantification. Three independent runs were performed for each sample. The results were expressed as nmol of carbonyls/mg protein.

### 2.8. Molecular Modeling of Pajuro-Derived Peptides Interaction with ACE-I and DPP-IV

Crystallographic data of the human ACE-I C-domain complexed with an enalaprilat inhibitor and human DPP-IV complexed with Diprotin A were downloaded from the PDB database (http://www.rcsb.org, accessed on 14 July 2022), under the codes (resolutions) 1UZE (1.82 Å) and 1NU8 (2.50 Å), respectively [31,32]. The enalaprilat inhibitor, which closely resembles the FAP sequence, and diprotin molecules, which have a peptide nature, were used as templates for designing structure of pajuro peptides on the binding sites of ACE-I and DPP-IV by side-chain amino acids replacement, keeping the backbone conformation of original inhibitors, using YASARA 21.8.27 program [33]. Each obtained model for structure of complex between ACE-I and DPP-IV proteins and pajuro peptides was refined by energy minimization, followed by a 500 ps of molecular dynamics (MD) simulation, utilizing the YASARA macro called md_refine for enhancement of built model. Previous refinement, each complex model was included in a box filled with solvation water molecules, 0.9% of NaCl ions, and considering NVT canonical ensemble. The pH used to define ionization states of amino acid residues was 7.4, the temperature was 298 K, and the density was 0.997. The pKa was computed for each residue according to the Ewald method [34]. The energy minimization of the model was performed with combined steepest descent and simulated annealing by attaching the backbone atoms of the aligned residues to avoid potential destruction of the model. During MD simulation 20 snapshots were saved every 25 ps; after calculus, the minimum energy model was selected for its structural analysis.

In order to identify amino acid residues importance for binding of studied enzymes to pajuro peptides and their reference inhibitors, a residue-based contribution analysis by in silico alanine scanning on selected structures with the DRUGSCOREPPI web server based on a knowledge-based scoring function was performed [35]. Each selected structure in the previous step corresponding to ACE-I and DPP-IV were structurally aligned by MUSTANG in YASARA [36] for comparative purposes. The models were visualized with UCSF ChimeraX [37], and additional analyses for data processing were realized using in-house Python scripts.

### 2.9. Statistical Analysis

At least 3 independent assays were performed, and results were expressed as the mean ± standard deviation (SD). Data were analyzed using a one-way analysis of variance (ANOVA), followed by the Newman–Keuls test for multiple comparisons. All analyses were run with the program Sigma Plot 14.5; a *p*-value < 0.05 was considered as significant.

## 3. Results and Discussion

### 3.1. In Vitro Multifunctionality of Pajuro-Derived Peptides

Fourteen synthetic peptides derived from E. edulis protein were selected and synthesized to evaluate their in vitro antioxidant, anti-hypertensive, anti-diabetic, and anti-obesity activity. As shown in Table 1, all peptides presented antioxidant activity through a dual mechanism of action: non-competitive electron transfer (ABTS) and hydrogen atom transfer (ORAC). The TEAC values ranged from 0.11 to 1.18 µmol TE/µmol peptide, and the ORAC values from 0.82 to 3.83 µmol TE/µmol peptide. These values are similar to those recently reported for synthetic animal-protein-derived peptides [38].

The presence of tyrosine and/or tryptophane could determine the antioxidant activity of pajuro peptides, as these amino acids have been recognized as the main contributors to the ABTS and peroxyl radical scavenging properties of food-derived peptides [20,40]. Moreover, other factors such as the length of the peptide chain and the position of bioactive amino acids into the sequence could influence on the antioxidant activity [41]. Only, three of fourteen analyzed peptides showed ACE inhibitory activity, with values ranged from 50.50 to 115.60 µM (Table 1). The lowest IC_50_ value was determined for peptide SQLPGW. The presence of proline at the C-terminal tripeptide could determine its potent inhibitory effect as it has been demonstrated that ACE favors substrates/competitive inhibitors with hydrophobic amino acids and/or proline at the three C-terminal positions [42].

α-glucosidase and α-amylase are enzymes involved in plasma glucose regulation, inhibition of which lead to a reduction of the global absorption of glucose into the blood flow. α-amylase accelerates the hydrolysis of glycosidic bonds of carbohydrates and starch in food. Inhibitors of this enzyme can act reducing the hydrolysis of glycosidic bonds, decreasing the digestion rate of carbohydrates, prolonging the absorption of glucose, and reducing the post-prandial blood glucose levels [43]. The substrate-binding pocket of α-amylase, containing a high number of aromatic amino acids, can interact directly with the aromatic residues (phenylalanine, tryptophan, and tyrosine) of peptide substrates/inhibitors through hydrogen bonds, electrostatic, and Van der Waals connections. These interactions seem to be critically implicated in the mechanism of action of inhibitory peptides towards α-amylase [44]. Although different studies have demonstrated that food protein hydrolyzates can exert α-amylase inhibitory activity [45], the information about the peptide sequence responsible for the observed effects is much more limited. In our study, only three of fourteen peptides showed a slight inhibitory activity against α-amylase (10.6–32.1%) at the assayed concentration of 100 µM. This activity was higher than that shown by synthetic peptides identified in simulated gastrointestinal digests from quinoa proteins [46]. The highest inhibition was determined for peptide CCGDYY. This peptide contained in its sequence two tyrosine residues in addition of one glycine residue that has been reported to improve the inhibitory properties of peptides [47]. However, the potential contribution of other residues which implication has not been studied yet cannot be discarded.

The starch (polysaccharide) in food is digested by saliva and pancreatic amylase into oligosaccharides that enter the small intestine and are broken down by α-glucosidase located on the brush border of the jejunum epithelial cells [48]. The action of α-glucosidase inhibitors results in a reduction of the sugar absorption area and delay in the absorption time, which is beneficial for decreasing post-prandial hyperglycemia, and thus, is effective against diabetes [49]. Although sugar-mimetic compounds are recognized as the best α-glucosidase inhibitors [50], other compounds from different nature such as flavonoids and tannins have also been characterized by their potent inhibitory activity [51]. Moreover, in the last years, the great potential of food-derived peptides has been suggested due to their affinity and specificity on the molecular target [52,53]. Oseguera-Toledo et al. isolated hypoglycemic peptides from de-hulled hard-to-cook beans hydrolyzates, suggesting that fractions less than 1 kDa were the main responsible as they showed the highest α-glucosidase inhibitory rate, reaching to 76.4 ± 0.5% [54]. Vilcacundo et al. identified a quinoa-derived peptide, which sequence was IQAEGGLT, with potent α-glucosidase inhibitory capacity (IC_50_ value of 109.5 µM) [46]. Recently, Feng et al. purified and identified three peptides with α-glucosidase inhibitory activity from defatted camellia seed cake [55]. Although the accurate mechanism by which peptides could inhibit α-glucosidase is still unknown, it has been suggested that interactions with the enzyme’s active site through hydrophobic residues might be involved as it has been reported for other inhibitors [56]. In our study, peptides CCGDYY, MFTGPY, and GSYHDSK showed moderate inhibition with percentage values of 25.1, 15.8, and 19.6%, respectively, at 100 µM (Table 1).

The enzyme DPP-IV is another important therapeutic target that has attracted attention in the last years. It is a serine protease widely distributed in the body that performs different biological functions, including co-stimulatory functions, receptor activity, and interaction with different proteins [57]. Inhibitors of this enzyme have become a new class of drugs used to treat type 2 diabetes by inactivating glucagon-like peptide 1 (GLP-1) and glucose-dependent insulin, stimulating glucose-dependent insulinotropic polypeptide (GIP) [58], promoting the release of insulin, and thus, lowering blood sugar levels. Studies have shown that hydrolyzates from different animal and plant proteins are promising sources of DPP-IV inhibitors [45,59,60]. DPP-IV inhibitory peptides tend to have a hydrophobic character, short size (2–8 amino acid residues), and a proline residue located at the first, second, third, or fourth N-terminal position [61]. Additionally, Vilcacundo et al. reported that potent DPP-IV inhibitory peptides generally contain a branched-chain amino acid (leucine, isoleucine, or valine) or an aromatic residue (primarily tryptophan) at their N-terminal position [46]. Some of these structural attributes could determine the activity determined for pajuro-derived peptides. Thus, peptide YPSY showed potent inhibitory effects with an IC_50_ value of 32.6 µM, while peptides AALWE and MFTGPY showed moderate IC_50_ values of 130.6 and 214.3, respectively (Table 1).

Pancreatic lipase is a key enzyme implied in the intestinal digestion of triacylglycerols, major source of dietary calories. Inhibition of this enzyme results in decrease of fat digestion and absorption, and subsequently in moderate long-term decreases of body weight [62]. Thus, lipase inhibitors are considered as useful in the management of obesity [63]. Recent studies have reported inhibitory effects on lipase for alcalase hydrolyzates from pea and black bean proteins [64,65], although the peptide sequences responsible for the effects were not identified. In our study, six of fourteen pajuro-derived peptides showed inhibitory activity against pancreatic lipase at the assayed concentration of 100 µM with percentage values between 11.6 and 20.9% (Table 1). Because of the potential anti-obesity activity and antioxidant, anti-hypertensive, and anti-diabetic properties shown by these six peptides, they were selected to evaluate the protective effects in a cell model.

### 3.2. Neuroprotective Effects of Pajuro-Derived Peptides

The high production of free radicals in the brain together the accumulation of transition metals has been described as an important contributor to neuronal cell death [66]. Therefore, scavenging free radicals could inhibit neurodegenerative diseases [67]. Since the use of current therapies against neurodegenerative disorders has been associated with multiple adverse side effects, the interest has focused on naturally derived compounds with antioxidant activity that could exert neuroprotective effects [68]. In our previous study, a pajuro protein hydrolyzate, after sequential action of pepsin, trypsin, and alcalase, showed protective effects on neuroblastoma SH-SY5Y cells challenged by FeSO_4_ [28]. Thus, this model was used to evaluate the contribution of isolated peptides on the neuroprotective activity demonstrated for pajuro hydrolyzate. The selection of peptides was done on basis of their ABTS and peroxyl scavenging activity. Thus, six of fourteen peptides, with sequences DGLGYY, AALWE, MFTGPY, CCGDYY, SQLPGW, and YPSY, were proven in the cell model. First, the cytotoxicity of peptides in cells under basal conditions was evaluated. None of them, at the highest concentration assayed (100 µM), affected the cell viability (data not shown).

Oxidative damage provokes an increase of intracellular ROS, which leads to cell death. Therefore, we investigated the protective effects of peptides against the oxidative damage induced by FeSO_4_ through quantification of the intracellular ROS levels using fluorescence with DCFA-DH. Cells were pre-incubated with peptides (25, 50, and 100 µM) for 3 h, and then, induced by FeSO_4_ (Figure 1A–F). In non-treated cells, FeSO_4_ induction resulted in a significant increase of ROS levels up to 158.5% (compared to non-induced cells). However, pre-treatment of cells with the peptides protected cells from the oxidative damage caused by FeSO_4_ in a dose-dependent manner. At the highest dose assayed (100 µM), the ROS levels were ranged between 113.7 and 128.6%, thus indicating an 18.7–28.7% reduction with respect to challenged cells. Although the data are still scarce, some food protein hydrolyzates from both animal and plant origin, and isolated peptides have been reported to reduce ROS levels in induced neuronal cells. Thus, peptide SGGY, identified in a walnut protein digest under simulated gastrointestinal conditions, protected SH-SY5Y cells against the oxidative damage induced by H_2_O_2_ [69]. Similarly, peptide fractions lower than 3 kDa, derived from beef protein hydrolyzates, reduced ROS levels in H_2_O_2_-challenged SH-SY5Y cells [70]. This fraction demonstrated to protect against oxidative damage through other mechanisms, such as inhibition of nitric oxide (NO) production and cell nuclei fragmentation, reduction of apoptosis, and increase of the mitochondrial membrane potential [71]. 

Thus, our study aimed at evaluating the effects of synthetic peptides through modulation of other biomarkers such as the lipidic peroxidation, LDH activity, and carbonyl compound levels (Figure 2). MDA is a stable degradation product of lipid peroxides that attacks unsaturated fatty acids in the cytomembrane and leads to cellular damage. Therefore, the degree of lipid peroxidation can be reflected through the measurement of MDA levels. As shown in Figure 2A, FeSO_4_-stressed SH-SY5Y cells showed 3.9-fold higher content of MDA in comparison with non-stressed cells (38.9 nmol/mg protein versus 10.1 nmol/mg protein). However, all pure peptides statistically down-regulated the induced MDA production, indicating their potential to attenuate ROS-mediated membrane damage. In the case of cells pre-treated with 100 µM of sequences MFTGPY, SQLPGW, YPSY, DGLGYY, and AALWE, and further challenged with FeSO_4_, the MDA values were statistically similar to those measured in non-induced cells. Similarly, peptides identified in an Alcalase soybean protein hydrolyzate down-regulated MDA levels in H_2_O_2_-induced Caco-2 cells [72]. If the cell membrane is damaged, LDH is released into the culture medium, thus, its analysis is considered an important index to detect cell death [73]. In our study, the induction of LDH activity by the chemical challenge (149.8% compared to non-induced cells) was also reverted by pajuro peptides (Figure 2B). In the case of the sequence SQLPGW, the reduction was similar to that exerted by 100 nM Trolox. Other peptides have also been reported to protect cell membrane integrity affected by chemical challenges. Thus, peptide KDHCH reduced LDH activity in H_2_O_2_-induced HepG2 cells [74]. When the level of ROS increases more than the antioxidant cell systems, it results in lipids, proteins, and DNA oxidations. The protein damage reflects the formation of protein carbonyls which levels increase in oxidized cells [75]. Untreated SH-SY5Y cells (negative control) exhibited a basal level of protein carbonyls due to the normal oxidative metabolism (Figure 2C). Addition of 200 µM FeSO_4_ remarkably induced production of protein carbonyl by 2-fold. However, pre-treatment of cells with peptides YPSY and SQLPGW significantly suppressed the formation of protein carbonyl compared to challenged cells, which represents the protective effects of peptides against protein peroxidation, as it has been previously reported for peptide fractions of Saccharomyces cerevisiae [76].

### 3.3. Molecular Modeling of Pajuro-Derived Peptides Interaction with ACE-I and DPP-IV

The discovery of new inhibitors is based on the knowledge of the structural requirements for their inhibitory activity against a particular target. Thus, the application of computational tools at the molecular level combined with experimental practices could generate new evidence in this area [77]. ACE-I and DPP-IV are crucial therapeutic targets for the treatment of hypertension and diabetes, respectively. Because of the adverse effects associated to the use of synthetic inhibitors of both enzymes, the search is focused on natural inhibitors, especially food-derived peptides, without side effects [78]. The success of this search is based on results obtained from both experimental and computational works [79,80,81,82]. Thus, recent computational approaches have allowed identifying food peptides with competitive inhibitory activity against ACE-I [83,84,85]. ACE-I inhibitors can form a stable complex with the enzyme, impeding the access of ACE substrates such as angiotensin I to the active site through H-bonds and hydrophobic networks [86]. It is known that for potent inhibitors, carbonyls groups, together with _383_H, _387_H, and _411_E, interact with Zn^2+^ in a distorted tetrahedral geometry, altering the active site of the enzyme [86,87,88]. This behavior was found after the in silico alanine scanning of complexes between ACE-I and pajuro-derived peptides (SQLPGW, DGLGYY, and YPSY). These amino acid residues showed the best ΔG_binding_ contributions (Figure 3B,C). The ΔG_binding_ contribution was higher for fragment SQLPGW in comparison with the rest of analyzed peptides (Figure 3A–C). This fact could explain the lower IC_50_ value determined for this peptide. ACE inhibition is also influenced by the peptide composition. Thus, potent ACE inhibitors such as captopril and enalaprilat contain proline as C-terminal residue, whose pyrrolidine ring could interact closely with a cluster of aromatic residues such as _457_F, _527_F, and _523_Y [86,88,89]. The peptide binding could be enhanced for the inclusion of other amino acids with hydrophobic properties capable of interacting with the ACE-I S1 subsite rich in hydrophobic amino acids such as _354_A, _518_V, and _512_F [32,86,89].

In the case of DPP-IV inhibitory peptides, they have arisen from a wide range of natural sources [79,82,90]. Diprotin A, primarily identified in culture filtrates of Bacillus cereus, is one of the most potent DPP-IV inhibitory peptides reported to date [91]. On basis of the reported crystalline structure of this peptide complexed with DPP-IV [31], pajuro-protein-derived peptides with potential DPP-IV inhibitory properties were selected. Binding of competitive inhibitors to DPP-IV has been shown to involve several subsites on its protein structure. One of these subsites consists of a hydrophobic S1 pocket with a narrow structure, which comprise _662_Y, _630_S, and _666_Y residues [92]. In the interaction models between DPP-IV and pajuro peptides (YPSY, AALWE, and MFTGPY), all these residues have important ΔG_binding_ contributions (Figure 4B,C). On the other hand, the DDP-IV subsite S2, larger than the S1 and containing glutamic acid in positions _205_E and _206_E, binds inhibitors through the formation of salt bridges that are recognized as crucial for the inhibitory activity. In all the obtained models, these residues have important ΔG_binding_ contributions (Figure 4B,C), supporting the DPP-IV inhibitory activity shown by these peptides (Table 1). The fragment YPSY exerted the highest inhibitory activity against DDP-IV (IC_50_ value of 32.60 µM) that could be due to the presence of a proline residue in its structure and its smaller size in comparison to the other pajuro peptides that facilitates its accommodation in the DPP-IV binding site. In the case of DPP-IV inhibitory peptides from other natural sources, they contain at least one proline residue within the sequence and mostly as second N-terminal residue [61,90]. The diprotin A inhibitor, with a sequence of IPI, shows an IC50 value of 24.7 μM. It has been demonstrated that the proline residue of this inhibitor interacts with the 662Y residue of DPP-IV [31], as it has been predicted in our study for peptides YPSY and MFTGPY. These interactions with 662Y residue have similar ΔG binding contributions in the case of pajuro peptides and diprotin A, according to in silico alanine scanning (Figure 4C).

## 4. Conclusions

In this study, the multifunctional properties of fourteen novel peptides identified in an alcalase hydrolyzate from pajuro seed proteins were confirmed. The synthetized peptides exhibited in vitro antioxidant activity through radical scavenging, ACE inhibitory activity, anti-diabetic effects through inhibition of α-amylase, α-glucosidase, and DPP-IV, and/or obesity modulatory ability by inhibition of pancreatic lipase. Molecular modeling studies have also predicted that some pajuro peptides could bind cavity of ACE and DPP-IV. Further evaluation using an FeSO_4_-induced brain SH-SY5Y cell model revealed that pure peptides protected cells against the oxidative damage enhancing cellular redox status and eliminating oxidative degradation products. These findings help clarify the relationship between peptide structure and multifunctional antioxidant/antihypertensive/anti-diabetic/anti-obesity activity and supports the potential application of pajuro protein hydrolyzates as novel ingredients for development of functional foods. However, the bioavailability of these peptides should be confirmed, and additional preclinical and clinical studies should be carried out to assess their in vivo health benefits reducing the risk of oxidative stress and metabolic-disfunction-associated diseases.

## Figures and Tables

**Figure 1 antioxidants-11-02101-f001:**
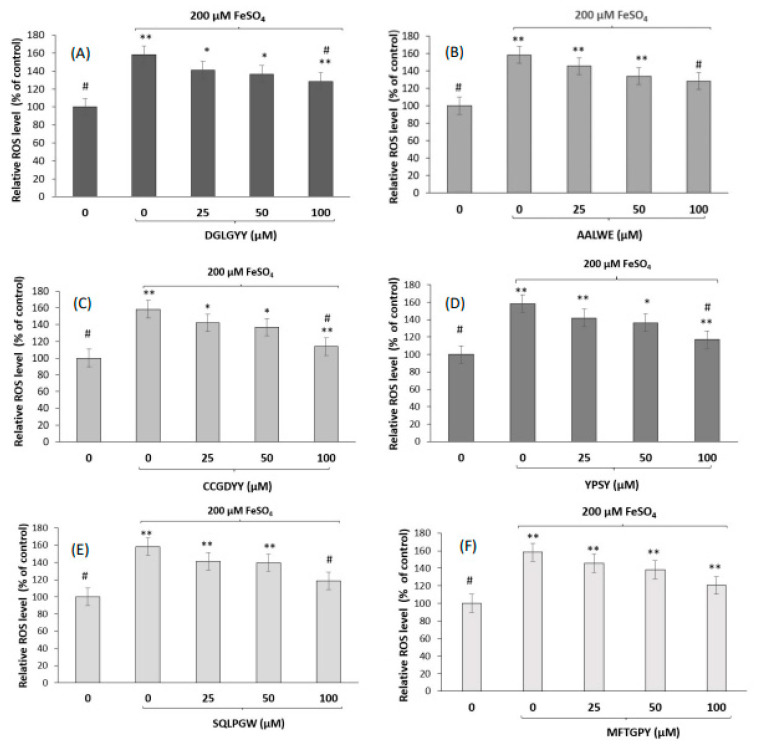
ROS levels (% of control) after the pre-treatment of SH-SY5Y cells with peptides (**A**) DGLGYY; (**B**) AALWE; (**C**) CCGDYY; (**D**) YPSY; (**E**) SQLPGW; (**F**) MFTGPY at different concentrations (25, 50, and 100 µM) for 3 h, and induction with FeSO_4_. * *p* < 0.05, ** *p* < 0.001 vs. control (non-treated cells), # *p* < 0.001 vs. FeSO_4_ treated cells.

**Figure 2 antioxidants-11-02101-f002:**
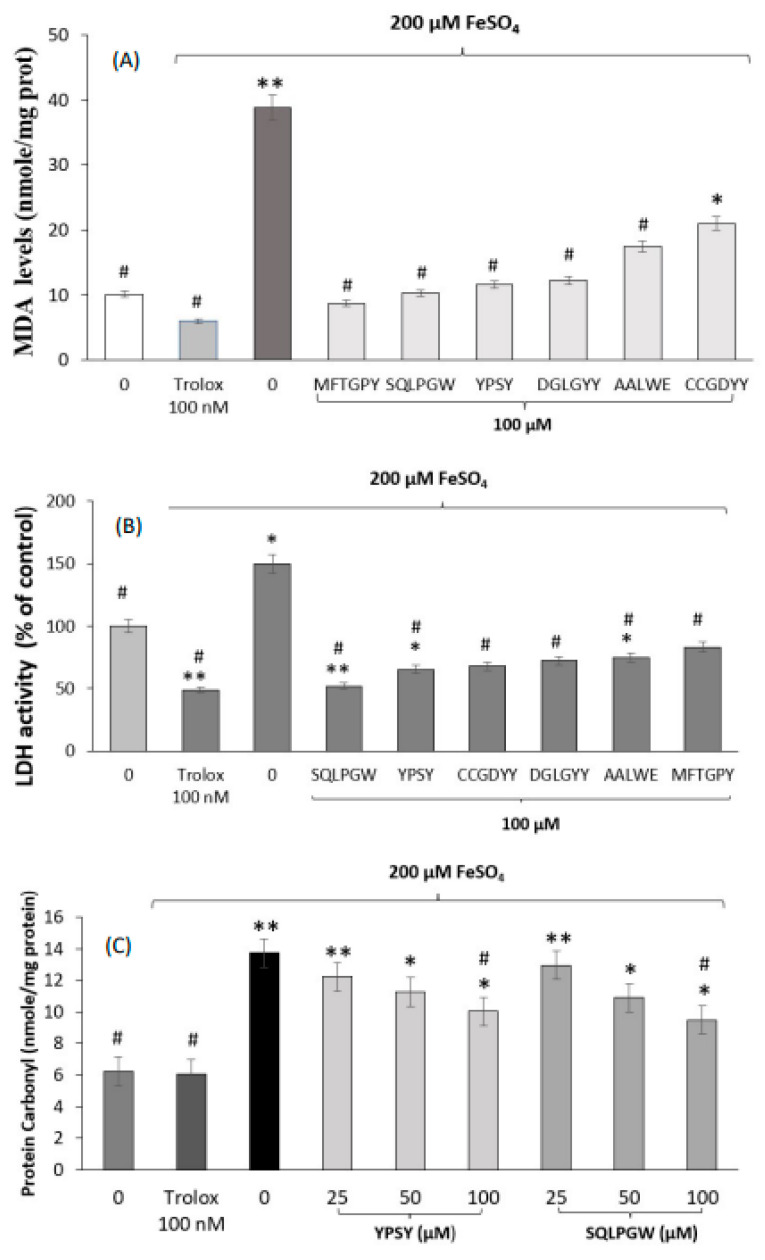
(**A**) MDA levels (nmol/mg protein) after the treatment of SH-SY5Y cells with FeSO_4_ in the absence and presence of 100 µM of pajuro peptides; (**B**) LDH activity (% of control) after the treatment of SH-SY5Y cells with FeSO_4_ in the absence and presence of 100 µM of pajuro peptides; (**C**) protein carbonyls levels (nmol/mg protein) after the treatment of SHSY5Y cells with FeSO_4_ in the absence and presence of YPSP and SQLPGW different concentrations (25, 50, and 100 uM). * *p* < 0.05 or ** *p* < 0.001 vs. control (non-treated cells), # *p* < 0.001 vs. FeSO_4_ treated cells.

**Figure 3 antioxidants-11-02101-f003:**
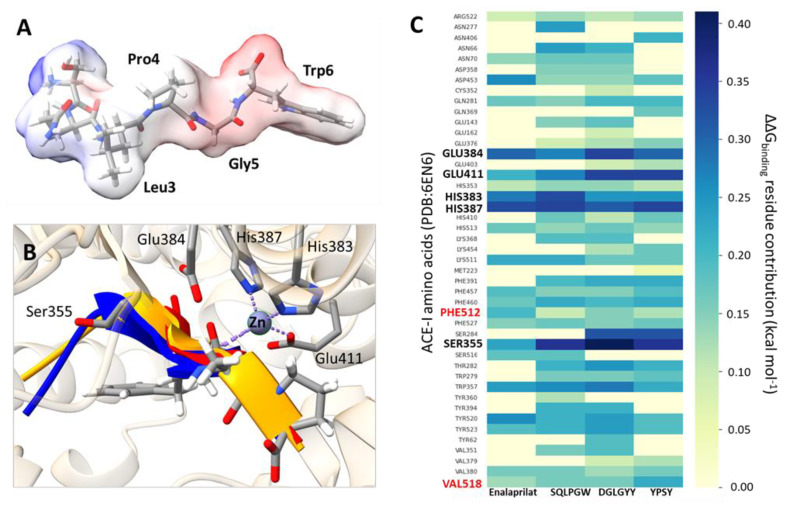
Molecular interactions between ACE-I and pajuro peptides after model structure refinement. (**A**) Representation of the SQLPGW peptide at physiological pH and its electrostatic potential isosurface showing negatively and positively charged regions in red and blue, respectively. (**B**) Side view (bottom) of the ACE-I pocket to illustrate the subsite occupancy by pajuro peptides. For pajuro peptides, only the backbone chain is shown, a chain with blue color corresponds to SQLPGW peptide, orange to DGLGYY peptide, and orange chain to YPSY peptide. Enalaprilat inhibitor atoms are shown explicitly in this representation. The figure also explicitly shows atoms of side chains from residues with higher contribution to ΔG binding according to in silico alanine scanning. (**C**) Heat map showing the ΔG binding contributions of ACE-I residues to their binding to pajuro peptides according to in silico alanine scanning using a DRUGSCOREPPI web server based on a knowledge-based scoring function. The name of residues with higher contributions are bold black, residues expected to have higher contributions in good ACE-I inhibitors are bold red.

**Figure 4 antioxidants-11-02101-f004:**
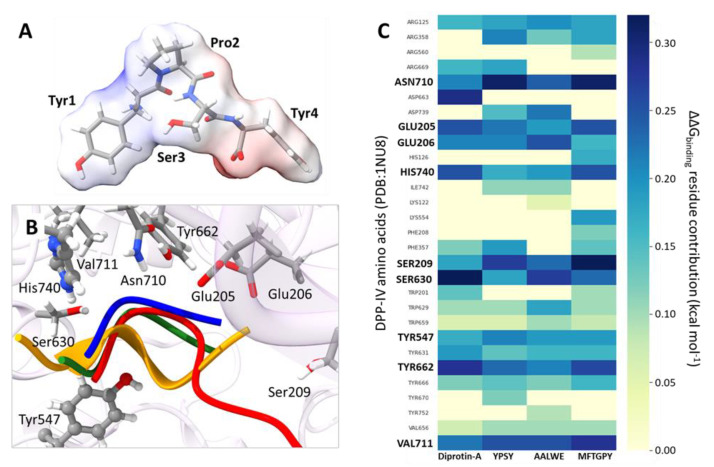
Molecular interactions between DPP-IV and pajuro peptides after model structure refinement. (**A**) Representation of the YPSY peptide at physiological pH and its electrostatic potential isosurface showing negatively and positively charged regions in red and blue, respectively. (**B**) Side view (bottom) of the DPP-IV pocket to illustrate the occupancy by pajuro peptides. For pajuro peptides, only the backbone chain is shown; chains with blue color correspond to YPSY peptide, orange to AALWE peptide, orange chains to MFTGPY peptide, and green chains to diprotin A inhibitor. The figure also explicitly shows side chains of residues with higher contribution to ΔG binding according to in silico alanine scanning. (**C**) Heat map showing the ΔG binding contributions of DPP-IV residues to their binding to pajuro peptides according to in silico alanine scanning using DRUGSCOREPPI web server based on a knowledge-based scoring function. The name of residues with higher contributions are bold black.

**Table 1 antioxidants-11-02101-t001:** In vitro antioxidant, anti-hypertensive, anti-diabetic, and anti-obesity activities of pajuro peptides characterized from Erythrina edulis seed.

Peptide	ABTS (μmol TE/μmol Péptide)	ORAC (μmol TE/μmol Peptide)	ACE Inhibition (IC_50_ = μM)	α-Amylase Inhibition ^a^ (%)	α-Glucosidase Inhibition ^a^ (%)	DPP-IV Inhibition (IC_50_ = μM)	Pancreatic lipase Inhibition ^a^ (%)
Control	*---*	*---*	6.81 ± 0.04	75.05 ± 1.55	62.36 ± 0.54	6.80 ± 0.03	94.62 ± 0.90
GPPW	0.26 ± 0.01	2.96 ± 0.38	n.d.	n.d	n.d.	n.d.	n.d
TWVV	0.11 ± 0.01	2.27 ± 0.07	n.d.	24.30 ± 0.01	n.d.	n.d.	n.d
YPSY	1.13 ± 0.06	3.26 ± 0.21	115.60 ± 0.26	n.d.	n.d.	32.60 ± 1.60	15.56 ± 1.35
AALWE	0.80 ± 0.50	1.05 ± 0.20	n.d.	n.d.	n.d.	130.60 ± 2.34	18.19 ± 0.98
YYLTR	0.40 ± 0.01 *	1.04 ± 0.07 *	n.d.	n.d	n.d.	n.d.	n.d
CCGDYY	1.18 ± 0.03 *	3.61 ± 0.00 *	n.d.	32.10 ± 0.01	25.12 ± 0.54	n.d.	17.06 ± 1.05
GESWCR	1.12 ± 0.02 *	2.43 ± 0.01 *	n.d.	n.d	n.d.	n.d.	n.d
DGLGYY	0.63 ± 0.04 *	3.83 ± 0.19 *	82.60 ± 0.58	n.d.	n.d.	n.d.	20.90 ± 1.31
MFTGPY	0.94 ± 0.01 *	2.44 ± 0.02 *	n.d.	n.d.	15.79 ± 0.55	214.30 ± 2.83	12.49 ± 2.01
SKDAPY	0.43 ± 0.04 *	1.06 ± 0.13 *	n.d.	n.d.	n.d.	n.d.	n.d
SQLPGW	0.53 ± 0.01 *	2.95 ± 0.24 *	50.50 ± 0.95	n.d.	n.d.	n.d.	11.63 ± 1.06
YDLHGY	0.64 ± 0.05 *	3.59 ± 0.46 *	n.d.	10.60 ±0.01	n.d.	n.d.	n.d
GSYHDSK	0.23 ± 0.02 *	1.74 ± 0.07 *	n.d.	n.d	19.57 ± 0.54	n.d.	n.d
NGENDWR	0.13 ± 0.01	0.82 ± 0.05	n.d.	n.d	n.d.	n.d.	n.d

*: ABTS and ORAC values reported by Intiquilla et al. [39]. ^a^: The percentage of inhibition of peptides was determined at 100 µM. n.d.: activity not detected. Control: trolox in ORAC and ABTS assays, captopril for ACE inhibition; acarbose (10 mM) for α-amylase and α-glucosidase activity; sitagliptin for DPP-IV inhibition; orlistat (0.5 µM) for pancreatic lipase inhibition. The values corresponded to the mean of three replicates ± standard deviation (DE).

## Data Availability

The data is contained within the article.

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
