# Peer review of "Protection against Oxidative Stress and Metabolic Alterations by Synthetic Peptides Derived from Erythrina edulis Seed Protein"

_antioxidants, 2022, doi:10.3390/antiox11112101_

Round 1

Reviewer 1 Report

This manuscript selects 14 antioxidant peptides derived from pajuro seed proteins, and finds effective peptides for hypertension and diabetes among them by in vitro screening. The results of predicting interactions between these peptides and ACE or DPPIV by computational approach are interesting because they are consistent with the actual inhibitory effects of these enzymes.

Major comments

1, The authors may select 14 peptides with antioxidant properties (TEAC, ORAC) from the pajuro peptides obtained by alcalase hydrolysis. Are there any possibilities that authors missing ACE and DPPIV inhibitory peptides without antioxidative properties? Also, state clearly what authors think about the relationship between ACE or DPPIV inhibition and antioxidant properties in peptides.

2, Since ACE and DPPIV exert their actions in the blood, inhibitory peptides must be absorbed into the blood. In the gastrointestinal tract, it is also degraded by digestive enzymes. These points are not considered at all in this study. The peptides in Table 1 may not or hardly be absorbed into the blood as the intact form. Authors should consider and mention these important aspects for peptides to exert their actions in the manuscript.

3, Line 108-109 and Table 1: Is it possible that these peptides are produced in the gastrointestinal tract after ingestion of Erythrina edulis? How much is it likely to generate these peptides in the intestine? This is important for inhibition of amylase, glucosidase and pancreatic lipase.

Minor comments

1, L229-230: Was this supernatant include FeSO4?

2, Figure 2C: Are there any inhibitory effects on protein carbonyl production other than the two peptides shown in this figure? Results for other peptides should also be stated in the text. 

3, Fugure 4C: Regarding DPPIV inhibition of MFTGPY, it is expected from the result of Fig.4C that IC50 value is equivalent to or stronger than YSPY, but the value shows a much high value. Can authors think of any reason for this?

Author Response

This manuscript selects 14 antioxidant peptides derived from pajuro seed proteins and finds effective peptides for hypertension and diabetes among them by in vitro screening. The results of predicting interactions between these peptides and ACE or DPPIV by computational approach are interesting because they are consistent with the actual inhibitory effects of these enzymes.

Major comments

  1. The authors may select 14 peptides with antioxidant properties (TEAC, ORAC) from the pajuro peptides obtained by alcalase hydrolysis. Are there any possibilities that authors missing ACE and DPPIV inhibitory peptides without antioxidative properties?

Answer: In a previous study carried out in our laboratory (Intiquilla et al., 2019), 30 peptides were identified in a pajuro seed protein hydrolyzate. From them, 14 peptides were selected according to their potential multifunctional activity by an in silico analysis (using the BIOPEP database). The present study aimed at confirming this activity by biochemical and cell culture assays and determine the potential contribution of these peptides on the observed beneficial effects. This information has been added (line 86).

  1. Also, state clearly what authors think about the relationship between ACE or DPPIV inhibition and antioxidant properties in peptides.

Answer: In relation to the pathophysiological process of hypertension, it has been described that angiotensin II is involved in alterations of the oxidative balance, stimulating the production of reactive oxygen species (ROS), which produces excessive oxidation of macromolecules, damaging cardiovascular structures. Also, an increase of ROS levels results from the chronic hyperglycemia associated to type-2 diabetes. An explanation of the involvement of ROS and oxidative stress on the pathophysiology of multiple NCDs is included into the introduction section (lines 47-55). Thus, peptides with ability to inhibit ACE and DPP-IV enzymes may also have a positive effect on the oxidative status of the body, and vice versa, antioxidant peptides may exert beneficial effects against hypertension and metabolic alterations. The sentence (lines 59-61) has been rewritten for a better understanding of this relationship.

  1. Since ACE and DPPIV exert their actions in the blood, inhibitory peptides must be absorbed into the blood. In the gastrointestinal tract, it is also degraded by digestive enzymes. These points are not considered at all in this study. The peptides in Table 1 may not or hardly be absorbed into the blood as the intact form. Authors should consider and mention these important aspects for peptides to exert their actions in the manuscript.

Answer: Human ACE is expressed strongly in many types of endothelial cells, especially in the capillaries of the lung, as well as in epithelial cells in the kidney, small intestine, and epididymis. The localization of ACE on the epithelial cells of the small intestine and the renal proximal tubule also suggests that peptides affecting these tissues may well serve as substrates. DPP-IV is an enzyme, present on epithelial and endothelial cells, and expressed in numerous tissues including the liver, gut, placenta, lung, and kidney. Since the multiple localization of both enzymes, peptides can act on them at both local and systemic levels. In the conclusion section (line 570), a sentence indicating that the bioavailability of peptides should be studied in addition of the preclinical and clinical studies to confirm their beneficial effects has been added.

  1. Line 108-109 and Table 1: Is it possible that these peptides are produced in the gastrointestinal tract after ingestion of Erythrina edulis? How much is it likely to generate these peptides in the intestine? This is important for inhibition of amylase, glucosidase and pancreatic lipase.

Answer: As it has been demonstrated for other food proteins, gastrointestinal digestion results in the release of bioactive peptides. Also, in the case of pajuro protein, a previous study carried out in out laboratory has recently demonstrated that the hydrolyzate obtained after sequential digestion with pepsin, trypsin and alcalase, showed antioxidant, and ACE and DPP-IV inhibitory activities (Palma-Albino et al., 2021). These findings suggest that some of the peptides analyzed in the present study could be released from pajuro protein after its passage through gastrointestinal tract and its digestion by digestive enzymes. An additional sentence has been added (lines 413-415).

Minor comments

  1. L229-230: Was this supernatant include FeSO4?

Answer: A correction in the materials and methods has been made (lines 230-233).

  1. Figure 2C: Are there any inhibitory effects on protein carbonyl production other than the two peptides shown in this figure? Results for other peptides should also be stated in the text.

Answer: Because peptides YPSY and SQLPGW were those that demonstrated the highest activity against other oxidative stress-associated biomarkers (LDH and MDA), the effects on protein carbonyl production was only determined for these two sequences.

  1. Figure 3C: Regarding DPPIV inhibition of MFTGPY, it is expected from the result of Fig.3C that IC50 value is equivalent to or stronger than YSPY, but the value shows a much high value. Can authors think of any reason for this?

Answer: As it is indicated (lines 521-522) in the case of peptide YPSY, the presence of proline at the penultimate position and its small size could be determinant for its high DPP-IV inhibitory activity. In the case of peptide MFTGPY, proline is also present at the penultimate position, but its size is bigger, thus, it could determine its higher IC50 value as DPP-IV inhibitor.

Reviewer 2 Report

In this study, the authors have examined the biological effects of synthetic peptides (with sequences identical to those derived from pajuro seed protein hydrolysis) focusing on anti-oxidant, anti-hypertensive And potentially beneficial metabolic actions. The comments on this manuscript are stated below:

1. The title and wording throughout this manuscript is quite misleading, since the authors tested synthetic analogues of the peptides selected from a pajuro seed protein hydrolysate in a previous study. This fact (i.e., these are synthetic peptides and NOT derived from pajuro seed protein) must be clearly mentioned in the title, Abstract, and Introduction. Similarly, the statement in line 86 must be corrected accordingly.

2. The pajuro protein used to identify these peptide sequences was from the seed of the plant. Hence, it must be referred to as "pajuro seed protein" and not just "pajuro protein" (which could be derived from leaf, fruit flesh etc. too).

3. Introduction: This section is too long, poorly organized, and has too much detail on oxidative stress. Please break it up into paragraphs, each focused on a separate topic such as oxidative stress, other relevant metabolic alterations, an introduction to the pajuro/ basul plant, and a brief overview of their previous work leading to the current study. 

3. Please provide a strong justification for using synthetic analogues of pajuro seed peptides, instead of using a pajuro seed hydrolysate (containing a mix of such peptides) for this study.

4. Statistics: It must be mentioned that p<0.05 is taken as significant

5. Since the studies are performed on in-vitro cell free systems or on cell culture models, any changes observed may not be directly linked to effects of diseases. As such, it is preferable to mention "Protection against oxidative stress and metabolic alterations" instead.

6. It is unclear from Table 1, why some but not other peptides were chosen. This must be clearly stated in the text as well as demonstrated in this Table

7. As the effects on metabolic parameters are quite modest (compared to the positive controls used in Table 1), a strong rationale must be used to justify further investigation of such properties of these synthetic peptides.

8. It is unclear why the authors chose to determine molecular interactions of certain peptides with ACE and DPP IV (given their modest level of inhibition) instead of studying the molecular mechanisms underlying the (much stronger) anti-oxidant effects. Please provide a discussion on how such anti-oxidant actions could be exerted, with additional studies performed if necessary.

Author Response

In this study, the authors have examined the biological effects of synthetic peptides (with sequences identical to those derived from pajuro seed protein hydrolysis) focusing on anti-oxidant, anti-hypertensive And potentially beneficial metabolic actions. The comments on this manuscript are stated below:

1. The title and wording throughout this manuscript is quite misleading, since the authors tested synthetic analogues of the peptides selected from a pajuro seed protein hydrolysate in a previous study. This fact (i.e., these are synthetic peptides and NOT derived from pajuro seed protein) must be clearly mentioned in the title, Abstract, and Introduction. Similarly, the statement in line 86 must be corrected accordingly.

Answer: As suggested by the reviewer, the title has been changed to: “Protection against oxidative stress and metabolic alterations by synthetic peptides derived from Erythrina edulis seed protein”. Accordingly, the abstract (line 17) and the introduction (line 86) have been modified.

2. The pajuro protein used to identify these peptide sequences was from the seed of the plant. Hence, it must be referred to as "pajuro seed protein" and not just "pajuro protein" (which could be derived from leaf, fruit flesh etc. too).

Answer: As suggested by the reviewer and indicated at the previous point, the title has been changed to: “Protection against oxidative stress and metabolic alterations by synthetic peptides derived from Erythrina edulis seed protein”. Accordingly, the abstract (line 17) and the introduction (line 86) have been modified.

3. Introduction: This section is too long, poorly organized, and has too much detail on oxidative stress. Please break it up into paragraphs, each focused on a separate topic such as oxidative stress, other relevant metabolic alterations, an introduction to the pajuro/ basul plant, and a brief overview of their previous work leading to the current study.

Answer: as suggested by the reviewer, the introduction has been broken up into several paragraphs according to the included information in each one.

4. Please provide a strong justification for using synthetic analogues of pajuro seed peptides, instead of using a pajuro seed hydrolysate (containing a mix of such peptides) for this study.

Answer: in previous studies carried out in our laboratory, hydrolyzates obtained from pajuro protein have been demonstrated to act as antioxidant, ACE, and DPP-IV inhibitors. Although peptides contained in these hydrolyzates were identified, the real contribution on the observed effects were not demonstrated. Thus, the aim of this study was to evaluate the effects attributed to peptides selected from those identified by an in silico analysis and elucidate their mechanisms of action. The use of pure synthetic peptides allows carrying out these studies avoiding possible interferences with other compounds present in the hydrolyzate. The objective of the study has been modified (lines 89-95).

5. Statistics: It must be mentioned that p<0.05 is taken as significant

Answer: As indicated by the reviewer, the sentence “P value < 0.05 was considered as significant” has been added (lines 303-304).

6. Since the studies are performed on in-vitro cell free systems or on cell culture models, any changes observed may not be directly linked to effects of diseases. As such, it is preferable to mention "Protection against oxidative stress and metabolic alterations" instead.

Answer: As suggested by the reviewer and previously indicated, the title has been changed to: “Protection against oxidative stress and metabolic alterations by synthetic peptides derived from Erythrina edulis seed protein”.

7. It is unclear from Table 1, why some but not other peptides were chosen. This must be clearly stated in the text as well as demonstrated in this Table

Answer: Peptides were selected according to the number of biological activities. At least, peptides should exert three activities. Six of the peptides acted as potential antioxidant and anti-obesity agents, and at least they exerted one additional activity (ACE inhibitory and/or anti-diabetic), thus they were chosen for the confirmation of the effects in the cell model. This explanation is included (lines 403-406).

8. As the effects on metabolic parameters are quite modest (compared to the positive controls used in Table 1), a strong rationale must be used to justify further investigation of such properties of these synthetic peptides.

Answer: We agree the reviewer that the inhibitory activity against enzymes involved in metabolic processes (DPP-IV, α-amylase, and α-glucosidase) is moderate (although comparable with peptides from other food sources as it has been explained into the results section, lines 349-351, 365-376, 383-385). However, as it has been described in the introduction section (lines 47-55), oxidative stress and hypertension are also involved in metabolic-alterations associated diseases, thus, those agents acting on different biomarkers such as multifunctional peptides, are considered as the best strategy to prevent/manage these disorders. These multifunctional peptides can act on different targets synergically, resulting in most potent effects.

9. It is unclear why the authors chose to determine molecular interactions of certain peptides with ACE and DPP IV (given their modest level of inhibition) instead of studying the molecular mechanisms underlying the (much stronger) anti-oxidant effects. Please provide a discussion on how such anti-oxidant actions could be exerted, with additional studies performed if necessary.

Answer: We agree the reviewer that the antioxidant effects of the peptides were more potent that the in vitro anti-hypertensive and anti-diabetic properties. In the case of ABTS and peroxyl radical scavenging peptides, the structure/activity relationship is already known, and it has been explained into the text (lines 324-329). According to our findings, the following objective was confirming the antioxidant activity demonstrated by the biochemical assays by using a cell model, specifically in neuroblastoma cells induced by FeSO4. In this model, the antioxidant activity was evaluated and the mechanisms of action on oxidative-stress associated biomarkers was elucidated. However, since the effects of peptides as ACE and DPP-IV inhibitors were only moderate, we preferred an in silico model to predict their ability to interact with both enzymes. 

Round 2

Reviewer 1 Report

The manuscript has been revised appropriately.

Reviewer 2 Report

The description "synthetic peptides derived from E. edulis seed protein" is incorrect, and should be replaced with "synthetic analogues of E. edulis seed peptides" or similar text.

Otherwise, there are no further comments.